# Experimental and Numerical Study of Static Behavior of Precast Segmental Hollow Bridge Piers

**DOI:** 10.3390/ma15196991

**Published:** 2022-10-09

**Authors:** Wenliang Lu, Wen-Qiang Peng, Li Zhu, Cong Gao, Ya-Dong Tang, Yue-Wu Zhou, Wei Su, Bing Zeng

**Affiliations:** 1School of Civil Engineering, Beijing Jiaotong University, Beijing 100044, China; 2Guangdong Provincial Academy of Building Research Group Co., Ltd., Guangzhou 510500, China; 3China Railway Design Corporation, Tianjin 300308, China; 4China Academy of Building Research, Beijing 100013, China

**Keywords:** concrete, precast segmental hollow bridge piers, static behavior, finite element analysis

## Abstract

To investigate the static performance of precast segmental hollow piers, two precast segmental hollow pier specimens were designed for static loading tests on the top of piers. The finite element model of precast segmental hollow piers was established by the finite element software Abaqus and verified based on the test results. Based on the experimental and finite element models, three optimal design solutions were proposed, and the calculation results of each solution were analyzed. The results show that precast segmental hollow pier mechanical behavior is similar to that of cantilevered bending members. The specimens present brittle damage characteristics after the destruction of the structure at the bottom of the pier pressure edge as the axis of the rigid body rotation. Following the test loading process, the bonding between the segments is good, except for the pier bottom damage surface of the rest of the bonding surface, which has no relative displacement. The calculation results of the finite element model are in good agreement with the test results and can effectively predict the load–displacement response of precast piers. Three optimized design solutions are proposed. The finite element simulation proves all three optimized design solutions show better overall ductility than the original solution and can effectively improve the performance of segmental precast hollow piers.

## 1. Introduction

In recent years, due to the dramatic increase in traffic volume, major cities have actively built additional elevated bridge expressways to relieve traffic congestion. However, in metropolitan areas, bridge construction can cause severe traffic delays and direct or indirect partial or total road closures. Accelerated bridge construction (ABC) technology offers higher safety, constructability, lower life-cycle costs, limited environmental impact, and better quality than traditional cast-in-place (CIP) construction methods [1]. A large number of scholars have conducted relevant and valuable studies, such as Farooq [2], Rad [3], Liu [4], Dániel [5], and Aksoylu [6,7,8,9,10,11]. Precast segmental piers, as part of the ABC technique, were first used in highway bridges in Texas in the 1970s and have been widely used in many low-earthquake zones [12,13,14].

Currently, many scholars have conducted experimental and theoretical studies on the mechanical properties of precast assembled piers. Nasir et al. [15] proposed a construction method for bridge piers using precast prestressed concrete slabs as formwork, and the feasibility of the method was verified by conducting tests on six different reinforced concrete column specimens. Hieber et al. [16,17] introduced a precast concrete system for rapid bridge construction, investigated the response of a segmental assembled double-column pier system using the OpenSees fiber model, proposed two segmental assembled double-column pier solutions from the construction point of view, and proposed a practical calculation method for maximum seismic displacement based on section cracking stiffness and base bottom shear strength ratio. Nishiyama et al. [18] simulated cyclic loading tests of posttensioned assembled column-foundation nodes under seismic loads and investigated the seismic performance of posttensioned assembled precast column-foundation nodes. Mander et al. [19] conducted an experimental study on unbonded posttensioned prestressed bridges with segmental joints at the base of piers for five different operating conditions to investigate their seismic performance as well as the reinforcement performance after damage. Billington et al. [20,21] proposed a precast segmental assembly system for the substructure of small- and medium-span bridges in non-seismic zones and established a planar unit model based on the DIANA finite element method to analyze the seismic performance of unbonded prestressed concrete piers. In addition, Billington et al. [22] suggested a precast segmental assembled pier system and proposed static tests on seven specimens to comparatively study the difference between the pier system with ductile fiber reinforced cement composite and the general concrete pier system. Hewes et al. [23] conducted a segmental assembled concrete sway system pier with four large-scale sizes connected by unadhesive prestressing tendons. YC et al. [24] carried out proposed static and dynamic tests on large-scale (1:2.5) hollow section segmental assembled piers to study prestressing reinforcement. Palermo et al. [25] conducted a proposed static test study of dry jointed segmental assembled piers with energy dissipating reinforcement and simulated dry jointed segmental assembled piers with energy dissipating reinforcement under cyclic loading using two parallel corner springs to simulate the moment-turn angle relationship at the base of the pier. This analysis method was widely validated and adopted by the New Zealand Concrete Design Code for the analytical design of this system of piers.

Kim et al. [26,27] conducted a static test study on four precast segmental piers set up with shear connection structures and precast foundations. Kurama et al. [28] studied the seismic construction of precast concrete wall slabs with energy dissipation devices embedded in longitudinal joints and transverse joints of the foundation connected with unbonded posttensioned prestressing bars, derived idealized trilinear base shear and top displacement display solutions under lateral loads, and proposed a seismic design method for shear walls of this system based on the current building seismic code. Mohamed [29] conducted numerical simulation and comparative verification of precast segmental assembled pier tests designed by different scholars based on ABAQUS/standard, and based on this, 84 precast assembled pier models with different design parameters were subjected to three-dimensional non-linear finite element analysis and design solutions, and calculation formulas for precast assembled piers were proposed.

In summary, the current research focuses on the seismic performance of segmental prefabricated assembled piers, and little research exists on the force behavior at the gluing surface of segmental prefabricated assembled hollow piers. In this paper, to study the force characteristics of segmental prefabricated hollow piers, two segmental prefabricated hollow pier specimens were designed and fabricated, with trapezoidal shear keys on the contact surface, connected by adhesive. The static tests were conducted on segmental prefabricated hollow pier specimens, and the finite element model (FEM) of segmental prefabricated hollow piers was established and verified based on the test results. Based on the experimental and finite element simulation results, we proposed three optimized design solutions to provide guidance for the design of segmental prefabricated hollow piers.

## 2. Experimental Program

### 2.1. Specimen Design

In this paper, two sections of prefabricated hollow pier specimens, S1 and S2, were are designed. The specimen overview is shown in Figure 1. The external structure of the specimen adopts uniform design dimensions, as shown in Figure 2. The pier section is a square hollow section, and the pier was evenly divided into three hollow sections of the same size. The section size is 400 mm × 400 mm × 350 mm, and the internal hollow size is 200 mm × 200 mm × 350 mm. The bearing size is 1300 mm × 1300 mm × 450 mm, and the bottom recess size is 1300 mm × 600 mm × 150 mm. The loading end size is 600 mm × 400 mm × 500 mm, and the top recess size is 1300 mm × 600 mm × 150 mm. The shape of the key is trapezoidal, its long side is 60 mm, and its short side is 40 mm high and 20 mm tall. The loading center is 250 mm from the top of the pier, the effective height of the pier column is 1750 mm, and the shear-span ratio is 4.38.

The experimental specimens were fabricated in the building materials laboratory and engineering structure laboratory of Beijing Jiaotong University. The fabrication process is shown in Figure 3. When assembling the prefabricated sections of the test specimen, it is necessary to use a brush to clean the floating dust at the splicing interface and then apply the uniformly blended adhesive to the bond. To ensure reliable bonding between the adhesive and concrete, after finishing the brushing and assembly of each section, the bolt fastening method was used to complete the extrusion work, as shown in Figure 3e. The test sections were connected to the lower bearing platform through 2 threaded thick reinforcement bars, which were tightened by manual application of force with nuts and spacers of corresponding specifications set at the top and bottom. During the tightening process, the phenomenon of colloid extrusion at the joints of the sections was observed, which indicates that a good extrusion effect was achieved by this method.

### 2.2. Material Properties

The concrete was self-mixed with ordinary silicate cement. Coarse aggregate was crushed stone with a grain size of 5~20 mm, fine aggregate was medium sand, and the water reducing agent was high-performance polyhydroxy acid water reducing agent. The proportion of concrete is shown in Table 1.

When casting each section, three 100 mm × 100 mm × 100 mm cubic test blocks and three 100 mm × 100 mm × 300 mm prismatic test blocks were cast, and the material performance test was carried out after curing for 28 days. The material properties of the concrete are shown in Table 2. The bonding between the sections is made of “CARBON” special adhesive for bridge section splicing, model CBJA, which is a two-component (A and B), solvent-free, modified epoxy, high thixotropic structural adhesive. The special adhesive can be used for bonding and sealing between bridge sections and meets all the requirements of CJJ/T 111-2006 [30] for adhesives used in bridge splicing construction, as well as the performance requirements of GB 50728-2011 [31] for structural class A adhesives. The A component is a white paste modulated by modified epoxy resin, thixotropic agent, additives, and fillers, and the B component is a black paste modulated by curing agent, thixotropic agent, additives, and fillers. The ratio of the two (quality) A:B = 2:1. The performance of bridge section splicing special adhesive is shown in Table 3.

The test piece of hoop reinforcement used a HPB300 steel bar with a diameter of 8 mm, and longitudinal reinforcement used a HRB355 steel bar with a diameter of 6 mm. Steel performance is shown in Table 4.

### 2.3. Test Setup

The test was carried out in the Engineering Structure Laboratory at Beijing Jiaotong University. The vertical loading equipment was a hydraulic jack with a specification of 1500 kN, and the horizontal loading equipment was an electrohydraulic servo actuator with a specification of 500 kN stroke ±250 mm of the MTS244 series. The specimen bearing platform is bolted to the horizontal floor, the hydraulic jack is placed vertically in the groove of the loading head, and the horizontal electrohydraulic servo actuator is connected to the loading head by bolts. The test loading is shown in Figure 4.

Before the official start of the test, the vertical load is prepressed, and then the vertical load is increased to 300 kN and kept constant. After completing the vertical load application, the horizontal loading equipment was controlled to carry out step-by-step loading. Through theoretical calculation, the displacement of the prefabricated hollow pier specimens’ stage is 6 mm when the peak bearing capacity is reached. Therefore, horizontal load was applied by displacement control, and the initial loading level is 1 mm. When it reached 6 mm, the loading level was adjusted upward by 2 mm increments. When it reached 12 mm, the loading level was adjusted upward by 4 mm. When it reached 28 mm, the loading level was adjusted upward by 6 mm. When it reached 40 mm, the loading level was adjusted upward by 10 mm. When it reached 50 mm, the loading level was adjusted upward by 20 mm. When it reached 70 mm, the loading test ended.

### 2.4. Instrumentation

The strain gauges were attached to the concrete surface of each section near the glue joint, and the patching scheme was symmetrical on both sides (forward and backward), with a total of 24 measurement points, as shown in Figure 5a. The steel strain gauges were pre-buried before specimen casting, with a total of 20 measurement points. The specific measurement point locations are shown in Figure 5b.

The vertical displacement sensor on the tensile side and the horizontal displacement sensor on the compressive side of the structure, with a total of 2 measurement points, were used to correct the errors caused by the movement of the bearing platform, as shown in Figure 5c.

## 3. Test Results

### 3.1. General Observations and Failure Modes

The test results show that the specimen S1 and S2 glue joints in the epoxy resin adhesive were not destroyed, the structure of the main crack was bonded to cut off, the bond residue was always glued to the prefabricated segments, the pressed area of concrete in addition to the stress concentration area at the bottom corner of the column was crushed, the rest of the pressed part did not exhibit an obvious pressure collapse phenomenon, and segment prefabricated hollow pier damage occurred, as shown in Figure 6.

The segment prefabricated hollow pier loading damage process is obvious and can be divided into three stages. When the loading level is less than 10 mm, specimens S1 and S2 do not exhibit visible cracks, the prefabricated pier sections do not show failure of the glue joint, and good integrity is maintained. When the loading level was 10 mm~24 mm, cracks began to appear at the bottom of the pier column and gradually developed from one end to the other end. As the load increases, the crack develops to the key position, and because of the stress concentration at the tip of the key position, a short crack develops obliquely along its tip, but the main crack in the structure still develops along its bottom to the pressurized concrete area of the cross-section of the pier column. Increasing the load, some of the tensile cracks develop obliquely upwards along the column body, and part of the concrete in the compressive zone is crushed, while the bearing capacity of the specimen gradually decreases. The whole member shows an obvious rigid body rotation. The crack at the bottom when the specimen is damaged is shown schematically in Figure 7.

### 3.2. Load–Deflection Curves

The full load–displacement curves of specimens S1 and S2 are shown in Figure 8. As Figure 8 shows, the whole process of stress damage of the specimen can be roughly divided into the elastic stress stage, elastoplastic stress stage, and brittle damage stage.

In the elastic stress stage, the elastic limits of specimens S1 and S2 are 1.5 mm and 2 mm, respectively, and the crack development at the bottom of the pier column shows the characteristics of brittle fracture because there is no tensile reinforcement at the cracking surface of the specimen. In the elastoplastic stressed stage, from the beginning of the crack at the bottom of the pier to the process of concrete crushing in the cross-sectional compression zone, test pieces S1 and S2 only had a 1~2 mm pier top displacement stroke. When the total displacement of the pier top reaches approximately 4 mm, the test pieces reach their bearing capacity limit and enter the brittle damage stage.

In summary, the segmental prefabricated hollow pier in the pier top horizontal thrust, before reaching its limit bearing capacity, exhibited force behavior similar to cantilevered bending members. The structure of the damage began in the pier bottom cross-sectional tension zone edge cracking. When close to the ultimate bearing capacity, the tensile crack at the bottom of the pier fully develops to the cross-sectional compression zone. After reaching the peak load, accompanied by a small sudden drop in the horizontal load on the top of the pier, the structure is transformed from the elastoplastic stress stage to an equilibrium system consisting of only the vertical force on the top of the pier, the gravity of the pier body, and the horizontal force on the top of the pier.

### 3.3. Load–Strain Response

The strain distribution of specimens S1 and S2 along the height direction of the section is shown in Figure 9. As Figure 9 shows, the strain changes in each section are in accordance with the assumption of a flat section, but because the most unfavorable cracking section causing structural damage occurs at the bottom of the pier and cannot be captured by strain measurement points, no obvious neutral axis rise appears in Figure 9.

The concrete surface strain–displacement curve at the top of the pier is shown in Figure 10. Figure 10 shows that the concrete surface strain of each section is consistent with the assumption of a flat section from the beginning of loading to the appearance of cracks at the bottom section of the pier. During the process of cracking at the bottom of the structure to reach the ultimate bearing capacity, the strains of measurement points 1−1 and 1−2 in the tension zone of the lower section decrease slightly and then remain stable, the compressive strains of measurement points 1−4 in the compression zone decrease, and the strains of measurement points in the remaining sections, except for the lower section, change slowly. When approaching the ultimate bearing capacity, the crack at the bottom of the pier is too large, resulting in the transformation of measurement points 1−3 from compressive to tensile strains.

After the load level exceeds the peak load, the compressive strain at the edge of the compression zone of the pier bottom section decreases, and the strain distribution of the rest of the cross-section is still in line with the assumption of a flat section. Analysis of the damage phenomena of specimen S1 and specimen S2 found that the column bottom sections of both specimens have a certain degree of bidirectional bending bias load phenomenon; that is, the neutral axis is not parallel to the rectangular edge of the column cross-section when the section is damaged.

## 4. Finite Element Analysis

### 4.1. Finite Element Model

#### 4.1.1. Finite Element Modeling

Based on the actual dimensions of the test members, the finite element analysis model of the segmental precast hollow piers was established by ABAQUS, and the explicit analysis module was used to display the dynamic analysis. In this model, the concrete is simulated by the 3D solid 8-node reduced integration unit C3D8R, the reinforcement is simulated by the 3D linear truss unit T3D2 and embedded in concrete, and the gluing joint between sections is simulated by the 3D 8-node cohesive element COH3D8. The finite element model of segmental prefabricated hollow piers is shown in Figure 11.

#### 4.1.2. Material Constitutive Relationship

The multiaxial behavior of concrete was modeled using the damaged plasticity (CDP) material model. The stress–strain relationship curves provided in Chinese concrete structure design code GB50010 [32] are used for the concrete compressive and tensile principal structures, as shown in Figure 12a. The constitutive relationship of concrete in compression can be expressed as:
(1)$$\sigma =\left\{ \begin{array}{l} E_{c}\varepsilon \frac{\rho_{c}n}{n-1+x^{n}}\;\;\;\;\;\;\;\;0\leq \varepsilon \leq \varepsilon_{c}\\ E_{c}\varepsilon \frac{\rho_{c}n}{\alpha_{c}(x-1{)}^{2}+x}\;\;\;\;\varepsilon_{c}<\varepsilon <\varepsilon_{cu} \end{array}\right.$$
where *α**_c_* = 1.39; *ρ**_c_* = *f_c_*/*E_c_*·*ε**_c_*; *n* = *E_c_*·*ε**_c_*/(*E_c_*·*ε**_c_*–*f_cu_*); *f_cu_* = 1.718 *f_c_*; *x* = *ε*/*ε**_c_*; *ε* is strain of concrete; *E_c_* is the tangential modulus of the elastic phase in compression; *f_c_* is the axial compressive strength; *ε**_c_* is the peak compressive strain; *ε**_cu_* is the ultimate compressive strain.

The constitutive relationship of concrete in tension can be expressed as:(2)$$\sigma =\left\{ \begin{array}{l} E_{c}\varepsilon \rho_{t}(1.2-0.2x^{5})\;\;\;\;\;\;\;\;0\leq \varepsilon \leq \varepsilon_{t}\\ E_{c}\varepsilon \frac{\rho_{c}}{\alpha_{t}(x-1{)}^{1.7}+x}\;\;\;\;\;\;\;\;\;\;\;\;\;\;\;\;\varepsilon_{t}<\varepsilon \end{array}\right.$$
where *α**_t_* = 1.39; *ρ**_t_* = *f_t_*/*E_c_*·*ε**_t_*; *x* = *ε*/*ε**_t_*; *ε* is strain of concrete; *E_c_* is the tangential modulus of the elastic phase in compression; *f_t_* is the axial tensile strength; *ε**_t_* is the peak tensile strain.

The principal structure of the steel beam with reinforcement is shown in Figure 12b, and the ideal elastic–plastic principal structure is as follows:(3)$$\sigma =\left\{ \begin{array}{l} E_{s}\varepsilon \;\;\;\;\;\;\;0\leq \varepsilon \leq \varepsilon_{y}\\ f_{sy}\;\;\;\;\;\;\;\;\;\varepsilon >\varepsilon_{y} \end{array}\right.$$
where *ε* is strain of steel; *E_s_* is the modulus of elasticity of steel; *f_sy_* is the yield strength of steel; *ε**_ψ_* is the yield strain of steel.

Contact surface are simulated using cohesive elements based on the maximum stress criterion to assign the damage evolution criterion with linear degradation of stiffness degradation to cohesive elements while ignoring the contribution of in-plane shear displacement and out-of-plane shear displacement to the fracture energy. The maximum stress at the beginning of the damage is taken as the actual tensile stress (*f_et_*) of the material, the fracture energy G*^I^_f_* varies from 40 N/m to 120 N/m for normal concrete to high-strength concrete, and G*^I^_f_* = 40 N/m is used conservatively in this paper. Stiffening is shown in Figure 12c.

The concrete tangential modulus of the elastic phase in compression (*E_c_*), axial compressive strength (*f_c_*), and axial tensile strength (*f_t_*) are from Table 2. The steel modulus of elasticity (*E_s_*) and yield strength are from Table 4. The tensile stress of cohesive elements is from Table 3.

### 4.2. Comparison of Test Results and FEM Analyses

Figure 13 shows the finite element simulation and test results. As show in Figure 13, the calculated curve peak and development trend are consistent with the test curve, and the damage process is consistent with the test, indicating that the modeling analysis method can effectively simulate the damage process of segmental prefabricated piers.

### 4.3. Optimizing the Design of Segmental Prefabricated Hollow Piers

#### 4.3.1. Optimization of the Design Scheme

Based on the test results, three optimized design plans are proposed. Plan A maintains the geometric configuration and reinforcement configuration of the original concrete section and adds two reinforcement bars of the same type as the original longitudinal bars between the top surface of the key projecting from the upper part of the section and the bottom surface of the key slot recessed from the lower part of the section. Plan B increases the height of the key position from 20 mm to 50 mm and adds two tensile bars of the same type as the original longitudinal bars between the top surface of the key position projecting from the upper part of the section and the bottom surface of the key slot recessed from the lower part of the section. In plan C, the raised keys are scattered from the middle of the cross-section to the periphery of the section to increase the moment of inertia of the reinforcement buried in the keys on the neutral axis of the cross-section, and four tensile bars of the same type as the original longitudinal bars are set inside each key position. The loading method in the finite element simulation is the same as the original test program, with a 300 kN vertical load applied to the top of the pier and kept constant, and the horizontal load is loaded by displacement control. The construction of the optimized design plans are shown in Figure 14.

#### 4.3.2. Analysis of Finite Element Simulation Results

The calculation results of each optimized design scheme are shown in Figure 15. As Figure 15 shows, the differences between the calculation results of the three optimized design solutions and the original solution are small until the vertical force is applied to the top of the pier and the tensile plastic strain appears at the bottom of the model pier. When the tensile plastic strain is applied at the bottom of the pier until the ultimate bearing capacity is reached, all three improvements appear to be better than the overall ductility of the test member. The distribution of tensile plastic strain during its destruction is scattered around the key due to the influence of the reinforcement, indicating that the improvement of the segmental pier structure is beneficial to the energy consumption of the external force during the destruction. The ultimate bearing capacities of plans A~C are 87.78 kN, 91.52 kN, and 93.28 kN, respectively.

Influenced by the increase in reinforcement at the key, the damage mode of the improved specimen is changed from brittle bending damage of plain concrete at the bottom of the pier to the bending damage mode of the reinforced concrete section, and its bearing capacity is also greatly improved compared with that of the test member. Comparing the improvement schemes A and B shows that the increase in the key height is beneficial to the increase in the section flexural bearing capacity. Comparing Option B and Option C shows that the key position is far away from the neutral axis of the section, which is beneficial to the stress of the reinforcement in the key position.

## 5. Conclusions

To study the force characteristics of prefabricated segmental assembled piers, this study designs and produces two segmented prefabricated assembled hollow pier specimens and conducts static loading tests on the top of the piers. In view of the shortcomings of the current design scheme, three optimized design schemes are proposed, and the calculation results of each scheme are compared and analyzed. Based on the experimental and finite element simulation results, the following conclusions can be drawn.

(1)The segmental prefabricated assembled hollow pier force behavior is similar to that of cantilevered bending members. The specimens present brittle damage characteristics after the destruction of the structure at the bottom of the pier pressure edge as the axis of the rigid body rotation.(2)During the test loading process, the bonding between the segments is good, except for the pier bottom damage surface of the rest of the bonding surface, which has no relative displacement. At the cracked damaged surface at the pier bottom, the connection bonds are always firmly connected to the key slot of the corresponding section.(3)The finite element model of precast segmental hollow piers was established, and the cohesive unit was used to simulate the adhesive joints between segments. The model calculation results are in good agreement with the test results and can effectively predict the load-displacement response of precast segmental hollow piers.(4)Three optimized design solutions are proposed. The finite element simulation proves all three optimized design solutions show better overall ductility than the original solution and can effectively improve the performance of segmental precast hollow piers.

## Figures and Tables

**Figure 1 materials-15-06991-f001:**
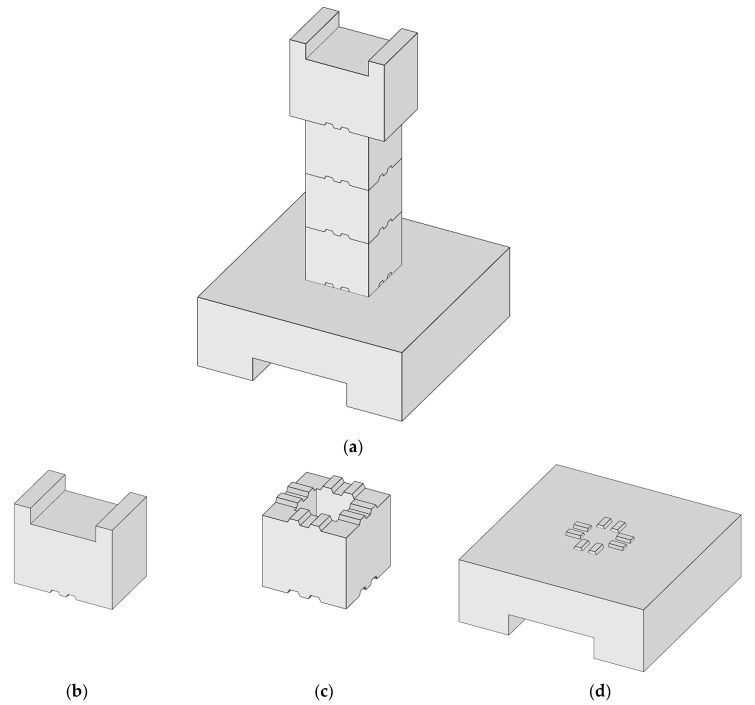
Specimen overview. (**a**) Segmental prefabricated bridge piers. (**b**) Load segment. (**c**) Segment. (**d**) Bearing platform.

**Figure 2 materials-15-06991-f002:**
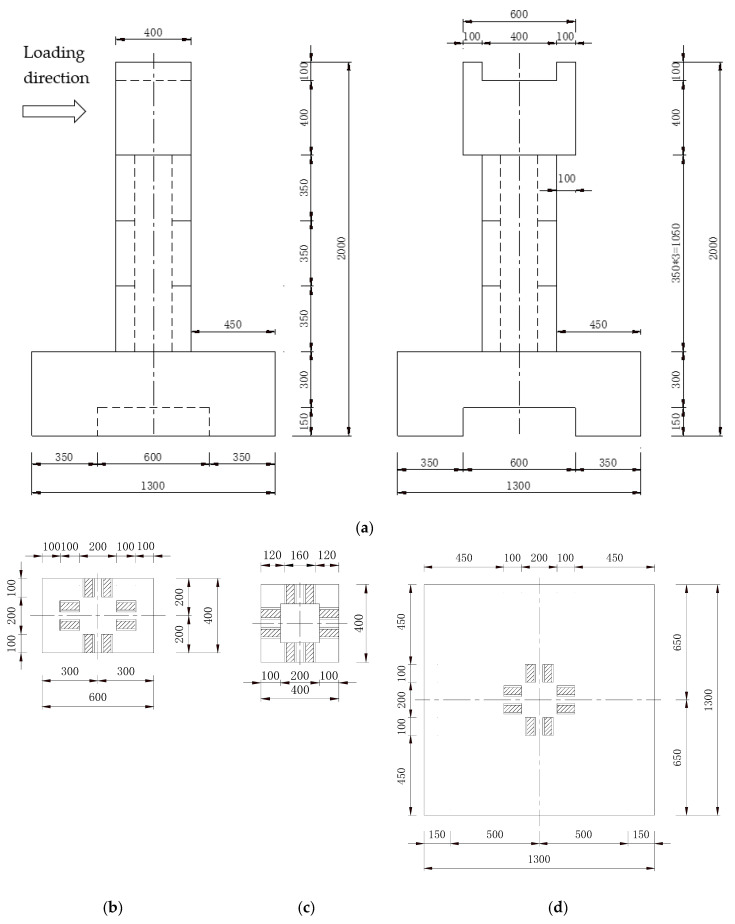
Details of specimens (units: mm). (**a**) Segmental prefabricated bridge piers. (**b**) Load segment. (**c**) Segment. (**d**) Bearing platform.

**Figure 3 materials-15-06991-f003:**
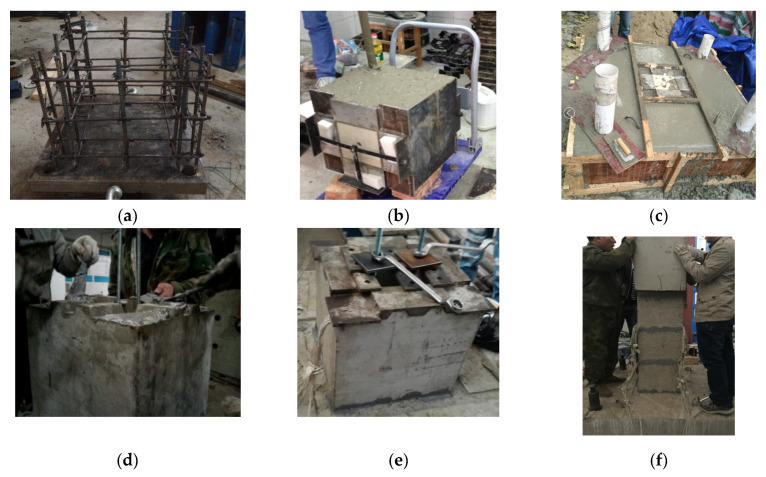
Fabrication process. (**a**) Tying of reinforcement cage. (**b**) Pouring sections. (**c**) Pouring bearing platform. (**d**) Contact surface gluing. (**e**) Extrusion bonding surface. (**f**) Section assembly.

**Figure 4 materials-15-06991-f004:**
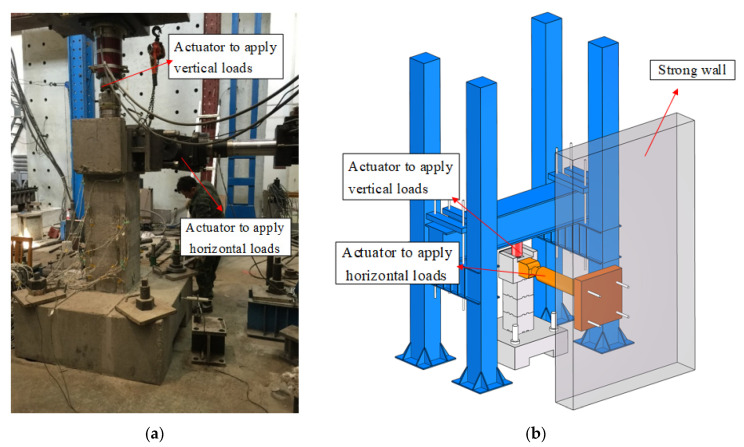
Test setup. (**a**) Photo. (**b**) 3D view.

**Figure 5 materials-15-06991-f005:**
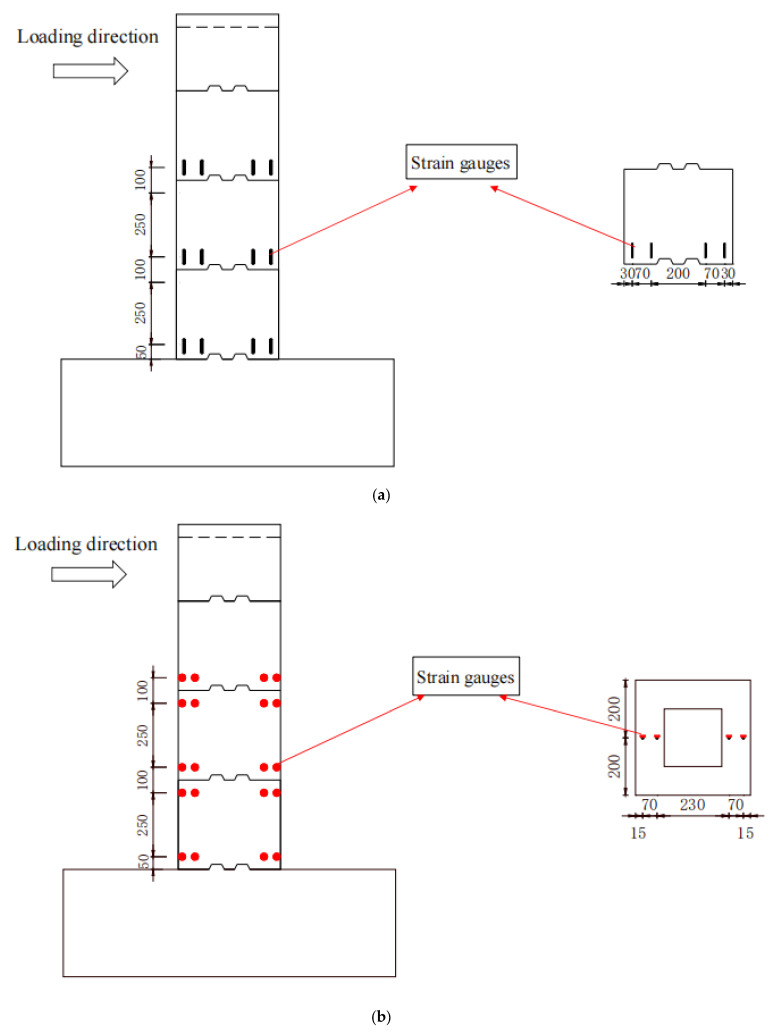
Instrumentation (in units of mm). (**a**) Strain gauges on concrete. (**b**) Strain gauges on reinforcing steel. (**c**) Displacement measure point.

**Figure 6 materials-15-06991-f006:**
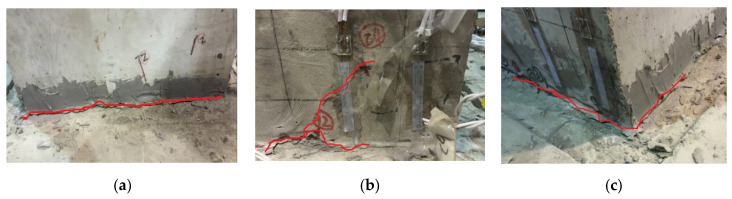
Failure mode of S1 and S2. (**a**) Pier bottom cracks of S1. (**b**) Oblique cracks of S1. (**c**) Pier bottom buckling of S1. (**d**) Pier bottom cracks of S2. (**e**) Penetration cracks of S2. (**f**) Pier bottom crushed of S2.

**Figure 7 materials-15-06991-f007:**
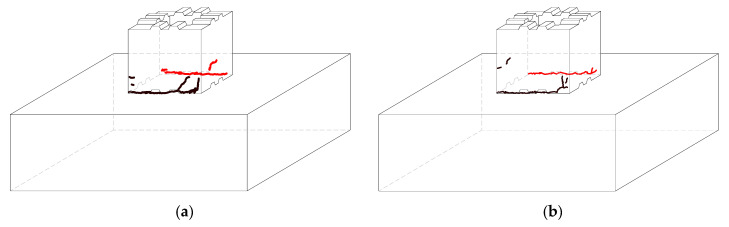
Crack pattern. (**a**) S1. (**b**) S2.

**Figure 8 materials-15-06991-f008:**
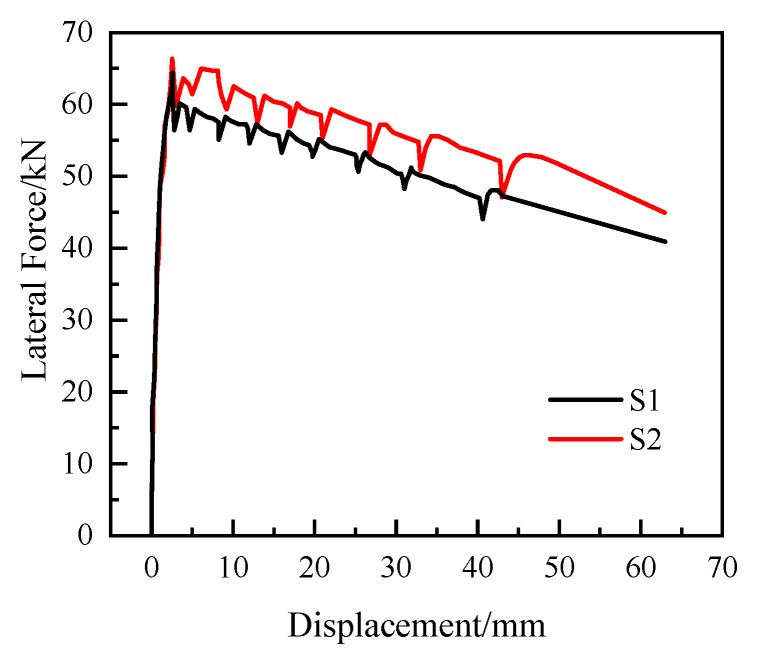
Load–deflection curves.

**Figure 9 materials-15-06991-f009:**
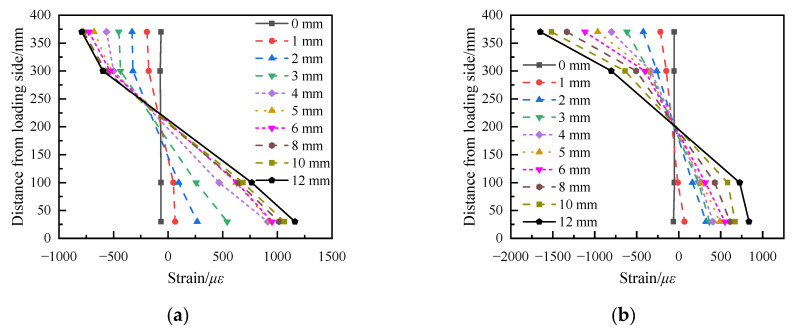
Strain distribution of each section. (**a**) S1 lower segment. (**b**) S2 lower segment. (**c**) S1 middle segment. (**d**) S2 middle segment. (**e**) S1 upper segment. (**f**) S2 upper segment.

**Figure 10 materials-15-06991-f010:**
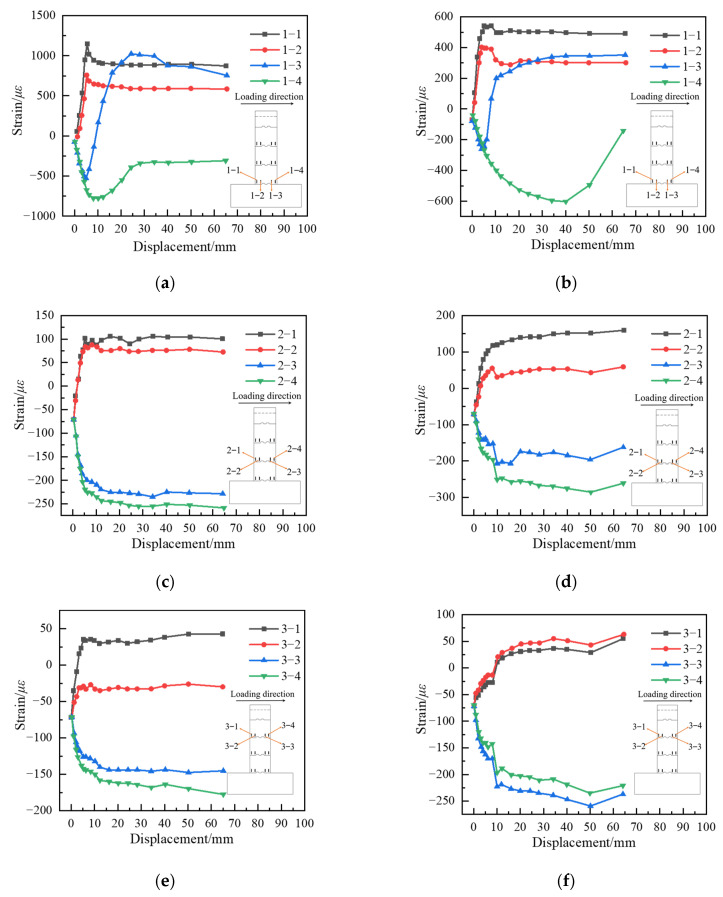
Displacement–strain curves for each section of specimen S1 and S2. (**a**) S1 lower segment. (**b**) S2 lower segment. (**c**) S1 middle segment. (**d**) S2 middle segment. (**e**) S1 upper segment. (**f**) S2 upper segment.

**Figure 11 materials-15-06991-f011:**
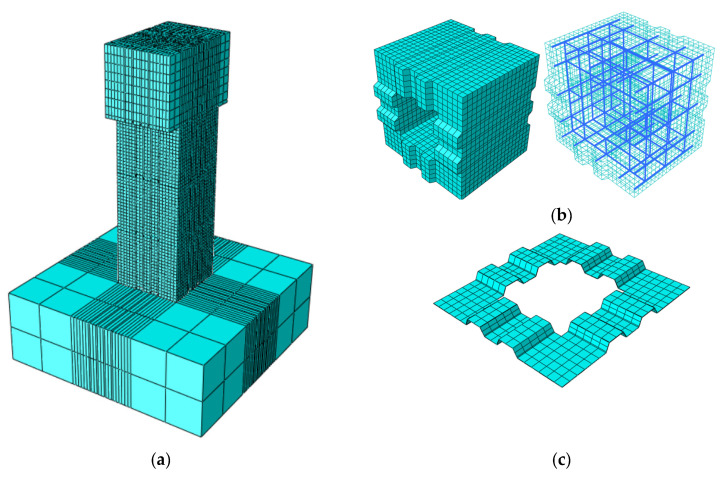
Modeling details of Specimen B2-2. (**a**) Segmental prefabricated bridge piers. (**b**) Segment. (**c**) Glue joint.

**Figure 12 materials-15-06991-f012:**
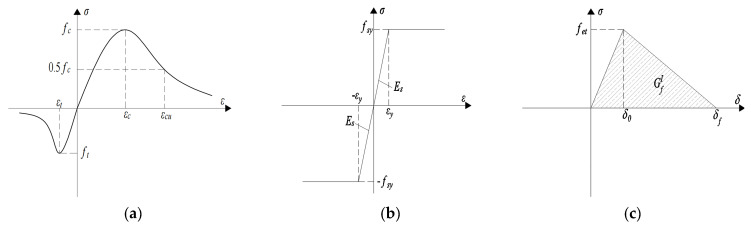
Material constitutive relationship. (**a**) NC. (**b**) Steel. (**c**) Cohesive.

**Figure 13 materials-15-06991-f013:**
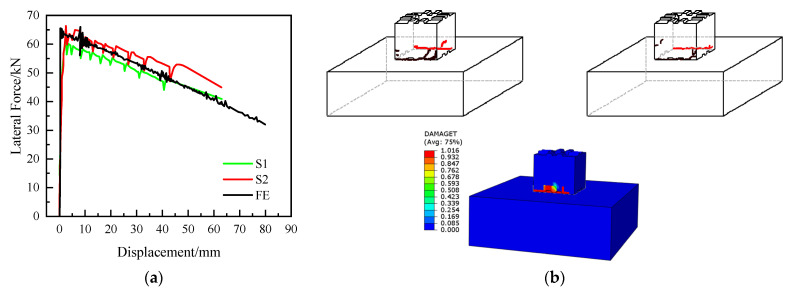
Comparison of the result between test and FEM simulation. (**a**) Load–deflection curve. (**b**) Damage pattern.

**Figure 14 materials-15-06991-f014:**
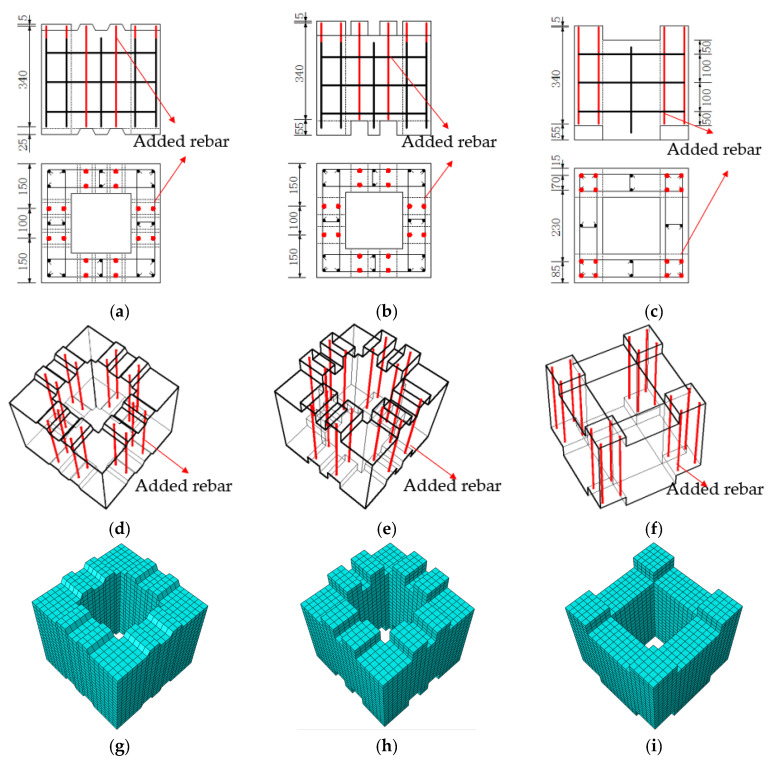
Optimized design solution. (**a**) Detail of plan A. (**b**) Detail of plan B. (**c**) Detail of plan C. (**d**) Segment overview of plan A. (**e**) Segment overview of plan B. (**f**) Segment overview of plan C. (**g**) Segment FEM of plan A. (**h**) Segment FEM of plan B. (**i**) Segment FEM of plan C.

**Figure 15 materials-15-06991-f015:**
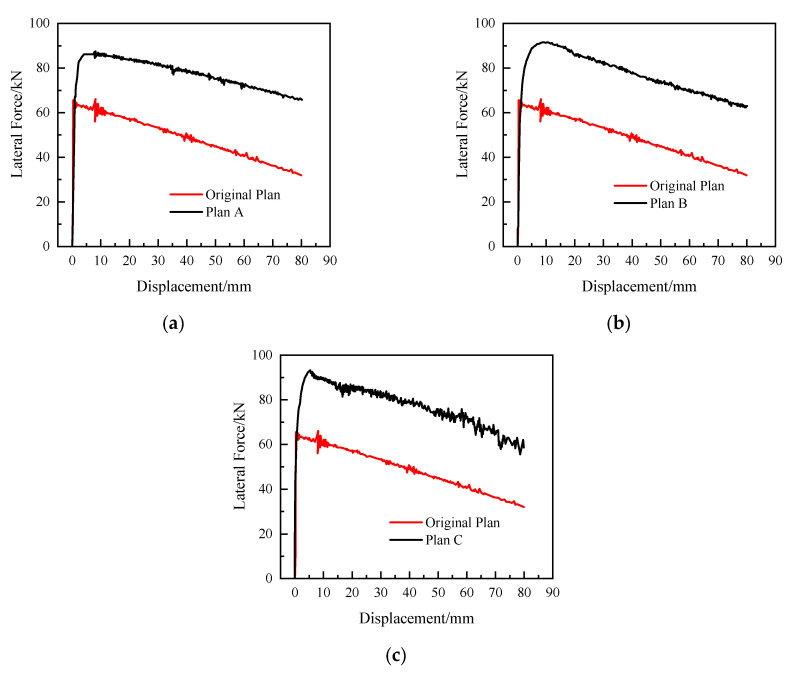
Comparison of the load–deflection curve of each scheme. (**a**) Plan A. (**b**) Plan B. (**c**) Plan C.

**Table 1 materials-15-06991-t001:** Composition of C50.

Material	Weight
	C50 (kg/m^3^)
Cement	478
Fine sand	610
Water	172
Superplasticizer	3.59
Coarse aggregate	1.186

**Table 2 materials-15-06991-t002:** Material properties of C50.

Mix	*f_c_* (Mpa)	*f_t_* (Mpa)	*E_c_* (Gpa)
C50	35.5	2.74	35.1

Note: *f_c_* is the axial compressive strength; *f_t_* is the axial tensile strength; *E_c_* is the modulus of elasticity.

**Table 3 materials-15-06991-t003:** Material properties of epoxy.

Materials	Compressive Stress (Mpa)	Tensile Strength (Mpa)	Elasticity Modulus (Gpa)
Epoxy	82	32	4.5

**Table 4 materials-15-06991-t004:** Material properties of steel.

Materials	*f_sy_* (Mpa)	*f_su_* (Mpa)	*E_s_* (Gpa)
HPB300	320	416	210.5
HRB335	345	511	203.2

Note: *f_sy_* is the yield strength; *f_su_* is the tensile strength; *E_s_* is the Young’s modulus.

## Data Availability

The data provided in this study could be released upon reasonable request.

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
