# Peer review of "Experimental and Numerical Study of Static Behavior of Precast Segmental Hollow Bridge Piers"

_materials, 2022, doi:10.3390/ma15196991_

Round 1

Reviewer 1 Report

The work presented an experimental and numerical study of static behavior of precast segmental hollow bridge piers. The results showed somehow that model can predict the load‒displacement response of precast segmental hollow piers. Besides, the proposed algorithm of optimization showed that the optimal solutions have better ductility than the original solution which means that the method can improve the force performance of segmental precast hollow piers. So far, the paper includes new contributions with good merits for publication. However, a number of issues/errors in the manuscript are expected to be solved.

 Required modifications:

-More details concerning the paper novelty, the aim of the paper and the results should be added to the introduction.

 - It will be more appropriate if you add a paragraph at the end of the introduction section illustrating the layout of the paper.

 -Finite element model section should be explained in more detail. Please explain the types of contact surfaces and steps in Abaqus model.

 - The introduction part needs to be extended by discussing more relevant papers. The authors should appropriately extend this section by discussing more relevant works focusing on different methods and models in the literature. For example, it is suggested to read and discuss the following relevant works:

Optimal Octagonal Hooked Collar Countermeasure to Reduce Scour Around a Single Bridge Pier”, Periodica Polytechnica Civil Engineering, 64(4), pp. 1026–1037, 2020. https://doi.org/10.3311/PPci.15966

Limit design of reinforced concrete haunched beams by the control of the residual plastic deformation, https://doi.org/10.1016/j.istruc.2022.03.080

Numerical study of confinement effectiveness in solid and hollow reinforced concrete bridge piers: Methodology, https://doi.org/10.1016/j.compstruc.2009.05.004

Reliability-based numerical analysis of glulam beams reinforced by CFRP plate, DOI:10.1038/s41598-022-17751-6

 -The authors should unite the style of figures (some of them are inside table and other figures are not ).

 -Please recheck the numbering of the figures ( Figure 5 is missing ).

 -There are some missing words from the descriptions which are within the figures.

 -In the equations, there are some expressions without definitions.

 -In section 2.2, the word CARBON is duplicated.

 -In section 4.1.2, please clarify what NC stand for.

Reviewer 2 Report

Congratulations to the author for the experimental study on prefabricated hollow bridge piers. 

I only have some minor editorial comments.

1. Heading 2 and sub heading 2.1 need to go to page no 5.

2. Figure 4 and 6 a) texts in the boxes are hidden and can not be read. 

Reviewer 3 Report

The authors conduct both experimental and numerical study to investigate precast hollow bridge piers. The results are interesting, there are many hardworking on this study. However, before acceptances there are major issues to be solved:

In abstract in the first line, instead of mechanical performance, more detailed words should be used.

In abstract, software ABAQUS should be stated.

The novelty of the papers can be presented in a better way to improve the quality of the manuscript.

More details can be provided for precast structures for example easy of application, connection details and etc.: experimental and numerical investigations of steel fiber reinforced concrete dapped-end purlins, experimental and numerical investigation of load bearing capacity of thinned end precast purlin beams and solution proposals, damages on prefabricated concrete dapped-end purlins due to snow loads and a novel reinforcement detail.

More detail should be provided abot CARBON. Why it is used? What makes it special?

Are they any loading protocol? What is the reason of applying, 2mm increments or 4mm_?

Load displacemnt curves should be provided in 1 figure

Why explicit modeling was used?

Did you model damage? If not, why the load displacement curve showed strength degradation?

No information was porvided for concrete parameters. What are the parameters used for this?

How did you model interaction between concrete and steel?

How did you select mesh size?

Finite element section can be detailed for further improve the reading quality and clearity. It is suggested to check these : nehavior of cfrp-strengthened RC beams with circular web openings in shear zones: numerical study, numerical evaluation of effects of shear span, stirrup spacing and angle of stirrup on reinforced concrete beam behaviour,numerical investigation of the parameters influencing the behavior of dapped end prefabricated concrete purlins with and without cfrp strengthening

Is it one model used for both S1 and S2?

No damaged photos of FE was provided. Please provide a picture with Exp vs Fe damage modes.

Fig 14 shows 3 plans. However, it seems that mesh size was different for each one. If these are mesh size, it will be irrevelant to compare them since mesh size can significantly impact the results

Did you do mesh size optimization?

What are the reason of fluctaions in the load displacement curves?

There are minor grammer mistakes which should be corrected

Reviewer 4 Report

The overall manuscript is good. The tests, simulation and result interpretation are inline. However, there are two spots for improvement. One is the abstract. Kindly try to be comprehensive and adherent to the objective of the manuscript. If possible reduce the contents of the abstract and rework. Second is the conclusion part. It is little descriptive in nature. If possible reduce the paragraphical length of the conclusions. Try not to describe the entire sequence again as observed in the tests and comparison. Rather, try to write the technical outcomes only.

Round 2

Reviewer 1 Report

I recommend the paper for publication.

Reviewer 3 Report

The paper can be accepted in this current form